# Assessing the Post-Brief-Storage Fruit Quality and Sensory Characteristics of Old, Local Apple Cultivars from the Carpathian Basin

**DOI:** 10.3390/plants14132005

**Published:** 2025-06-30

**Authors:** Gitta Ficzek, Sherif Mehmeti, Géza Bujdosó, Ágnes Magyar, Gergely Simon

**Affiliations:** 1Institute of Horticultural Science, Department of Fruit Science, Hungarian University of Agriculture and Life Sciences (MATE), 29-43. Villányi Str., 1118 Budapest, Hungary; ficzek.gitta@uni-mate.hu (G.F.); sherif.mehmeti@hotmail.com (S.M.); magyar.agnes.monika@uni-mate.hu (Á.M.);; 2Institute of Horticultural Science, Research Centre for Fruit Groweing, Hungarian University of Agriculture and Life Sciences (MATE), 2. Park Str., 1223 Budapest, Hungary

**Keywords:** antioxidant parameters, biological active compounds, breeding material, physical parameters, consumer evaluation

## Abstract

Local apple (*Malus x domestica* Borkh.) cultivars with tolerance to environmental stress factors must be re-evaluated. While the cultivation of apple trees has a long-standing tradition in Hungary, only a handful of cultivars are produced on a large scale, reflecting a trend in global apple production. The most commonly cultivated apple cultivars worldwide include ‘Golden Delicious’, ‘Red Delicious’, ‘Gala’, ‘Fuji’, and ‘Granny Smith’ (with ‘Jonagold’ and ‘Idared’ being significant in Europe). As a result, genetic diversity among apple cultivars has decreased significantly, which has increased the risk of epidemics if a new pathogen appears. Nonetheless, old and local apple cultivars of the Carpathian Basin have adapted well to Hungarian environmental stress factors and pathogens and seem tolerant to them. In this study, fruit analyses and consumer evaluations of eighteen old, local apple cultivars were conducted. Various physicochemical parameters, such as fruit mass, firmness, total soluble solid content, and total acid content, as well as the content of some biological active compounds, including polyphenol content, antioxidant capacity (FRAP), and pectin content, were determined. Additionally, a consumer evaluation was carried out. Based on the results, most of the old, local apple cultivars exhibit high fruit quality and offer considerable health benefits. The proportion of biologically active compounds in these cultivars is equal to or higher than that in the reference cultivar ‘Watson Jonathan’. Based on the excellent fruit quality and consumer preferences, the ‘Harang alma’ (an attractive fruit exhibiting high FRAP values) and ‘Marosszéki piros’ (with firm flesh and a high pectin content and being particularly tasty) cultivars are recommended for backyard gardens.

## 1. Introduction

In recent centuries, the Carpathian Basin has been characterized by an extreme diversity and richness of apple cultivars based on some pomological studies [1,2,3]. The old apple cultivars have constantly been pushed out of plant cultivation in the past few decades, and only apple cultivars that provide competitive market quality are integrated into intensive cultivation. Apple cultivars such as ‘Fuji’, ‘Granny Smith’, ‘Golden Delicious’, ‘Red Delicious’, ‘Idared’, and ‘Gala’ have become dominant in global apple production [4]. However, Hungarian apple cultivation is also characterized by the prevalence of ‘Golden Delicious’, ‘Red Delicious’, ‘Idared’, and ‘Gala’ apples. This has led to a drastic decrease in genetic diversity and poses a serious risk in the event of an epidemic [5,6]. The old apple cultivars have great value for the conservation of biodiversity, and many of them resist various diseases quite well [7,8,9,10]. The best solution to increasing the genetic diversity of apples is to collect and maintain accessions that derive from local populations or are well-adapted to local biotic and abiotic conditions [7,10]. According to Mihaljevic et al. (2021) [11], plants’ tolerance to various abiotic and biotic stresses, including drought, extreme temperature, disease, and pest resistance, is crucial in the context of global climate change. Such traits are typically found in traditional cultivars, but due to the widespread use of high-yielding commercial cultivars, traditional cultivars have become less popular. Since the Convention on Biological Diversity (CBD) adopted at the United Nations Conference on Environment and Development in Rio de Janeiro in 1992, the search for genetic resources and the collection and maintenance of old cultivars have been ongoing at an international level [4,9,10,12,13].

The world germplasm collections, planted to ensure genetic diversity, have become a primary focus in apple-producing regions [14,15,16,17,18,19,20,21,22,23,24,25,26].

In 1997, the Department of Fruit Growing of the Hungarian University of Agriculture and Life Sciences joined this initiative. The most important aim of this program is to find, collect, preserve, maintain, and evaluate the old apple cultivars in the Carpathian Basin [5]. A total of 120 old apple cultivars have been placed in the ex situ gene bank of the Department of Fruit Growing. Old apple cultivars with unique or distinctive flavors are sought by a growing number of customers. At the same time, the roles of fruits in health protection have been demonstrated in several studies [27,28].

Many papers have reported on the morphological and genetic variability of local-bred cultivars and their fruit quality [8,29,30,31,32,33,34,35,36]. Several studies have examined the relationship between gene-determined resistance and the number of antioxidants in apple cultivars [37,38]. According to Usenik et al. (2004) [39], the polyphenol concentration of the apple fruit determines the plant’s susceptibility to/resistance against *Venturia inaequalis* (CKE) Wint. For centuries, native apple cultivars have been adapted to the unique conditions of their local environments, resulting in the development of distinct genotypes, as described by Tetera (2006) [40].

The ‘Jonathan’ apple cultivar, along with its mutants, is popular in Central Europe among customers and the food industry due to its excellent fruit quality [5,7]. However, this cultivar is very susceptible to the three most important diseases (apple powdery mildew (*Podosphaera leucotricha* (Ell. et Everh.) Salm.), apple scab (*Venturia inaequalis* (CKE) Wint., and fire blight (*Erwinia amylovora* (Burrill) Winslow et al.)).

Several old cultivars have excellent health protection value, can be integrated into organic farming, have good resistance, and offer unique flavors [10,27].

Against the backdrop of current climate conditions and consumer preferences, evaluating the old cultivars is an important research activity. Old or historic cultivars have adapted well to local biotic and abiotic conditions over the course of several centuries. The aim of this study was to find some old varieties that offer good fruit quality and high biological activity and are suitable in terms of their actual market value.

## 2. Results and Discussion

### 2.1. Physicochemical Parameters

The physicochemical parameters of the old apple cultivars provided valuable insights into their unique characteristics and nutritional compositions. These characteristics were determined by the genetic backgrounds of the plant material, but they were also influenced by many factors, such as environmental conditions and cultural practices.

Upon comparing their physicochemical parameters, we found that many of the old cultivars reached the values of the control (Table 1). There were some exceptions: ‘Gyógyi piros’, ‘Herceg Battyán’, Simonffi piros’, and ‘Sóvári nobil’ had smaller fruits than the control. The following cultivars were classified as having larger fruits: ‘Batul’, ‘Marosszéki piros’, and ‘Bereczki Máté’. It is possible that the difference in average fruit mass reported in several papers [30,41,42,43] was due to varietal differences. Apple cultivars can have significant differences in terms of fruit mass, which can be influenced by genetic and environmental factors as well as cultural practices.

Fruit firmness is an important characteristic of apple cultivars from the points of view of customers and the food industry because of its association with post-harvest quality, shelf-life, and storability. This characteristic has a direct relationship with the juiciness and ripening status of fruits, but it has in inverse relationship with both previously mentioned characteristics [44]. According to the results reported by Cice et al. (2023) [41], the old Italian apple cultivars exhibit a firmness of 4.62 to 10.42 kg/cm^2^, which means that the firmness values of old apple varieties show a large degree of variability. The cultivars involved in our trial also reached this range.

The soluble solids content (SSC) provides important information about the sugar content of apples and their ripening statuses, information that is essential for determining harvest times [45]. The SSC of the examined apple cultivars varied between 12.4 and 17.4 °Brix. The refraction value of the ‘Máté Dénes’ cultivar was extremely high, while the extremely high sugar/acid content of the ‘Máté Dénes’ cultivar was due to its very low acid content. The results of studies conducted by many authors [31,41,42,46,47,48,49] are consistent with the presented findings regarding the total soluble solids content of old, local apple cultivars. These values are lower than the results for old cultivars grown in Poland (8.2–14.6 °Brix) [32]. In the case of old Italian cultivars (8.2–13.8 °Brix), our values are a bit higher [41]. Our results and the results from previous research show that old, locally bred apple cultivars show great variability in °Brix values.

The highest acid content was measured in the ‘Herceg Battyán’ apples, and the lowest was recorded for the ‘Máté Dénes’ apples. According to the results obtained by Oszmianski et al. (2018) [32], the total acid content of the 22 apple cultivars examined was between 0.17 and 1.07%, a range similar to that in our results. The findings of Jakubek et al. (2020) [49] indicated a relatively elevated level of titratable acidity ranging from 0.4 to 1.1%. In terms of the studies conducted by other research groups [31,42,47,50,51], the results were consistent with our findings regarding the titratable acid content of old, local apple cultivars. However, the total acid content of the cultivars in in our trial varied to a large degree; there were important differences between them in this regard.

The sugar/acid ratio is important for the taste of fruits. The refraction values for the ‘Máté Dénes’ and ’Herceg Battyán’ apples were extremely high, while the lowest value was found for ’Tükör alma’. The titratable acidity ranged from 0.22 to 1.1%. The greatest acid content was found in the ’Herceg Battyán’ apples, and the lowest was in the ‘Máté Dénes’ variety. The extremely high sugar/acid ratio of the ‘Máté Dénes’ apples was due to their extremely low acid content and high refraction values, leading to the predominantly sweet taste of this cultivar.

Based on our results, the examined varieties showed variability in SSC values and perceived sweetness, and the taste ranged from extremely sweet to very acidic.

These varieties should not be directly compared to commercial cultivars, as their morphological characteristics and flavor profiles differ significantly. Consequently, they are not suited for broad customer markets. However, niche customer segments with a preference for traditional sensory attributes often seek out these heirloom varieties due to their nostalgic value. Furthermore, these historical cultivars serve as valuable genetic resources for plant breeders, contributing to the preservation of biodiversity and the development of new varieties.

### 2.2. Biologically Active Compounds

The high proportions of biologically active compounds in the fruits of the old apple cultivars investigated were of great value (Table 2). Based on all the biologically active compounds examined, the fruits of the ‘Galambka’ and ‘Máté Dénes’ cultivars contained extremely high amounts of pectin and polyphenol and extremely high FRAP values. The fruits of the ‘Batul’, ‘Herceg Battyán’, ‘Kis Ernő Tábornok’, ‘Kisasszony alma’, ‘London pepin’, ‘Simonffi Piros’, and ‘Sóvári nobil’ cultivars also had valuable antioxidant sources. The fruits of the ‘Bereczky Máté’ cultivar had the lowest pectin and polyphenol content and FRAP.

Ficzek et al. (2017) and Oszmiański et al. (2018) [31,32] found comparable pectin content levels in old, local apple cultivars. The polyphenol content of these cultivars has been previously documented by Panzella et al. (2013), Lončarić et al. (2019), Piagentini et al. (2017), and Kschonsek et al. (2018) [52,53,54,55]. Overall, these studies suggested that traditional apple cultivars could be a rich source of polyphenols and that the polyphenol content of different cultivars could vary widely.

Our results suggest that old, locally grown apple cultivars represent a valuable source of FRAP, indicating their high antioxidant capacity. This finding is supported by several studies [56,57]. These studies reported similar levels of FRAP in apple cultivars, highlighting the potential health benefits associated with consuming these apples. The findings of Morresi et al. (2018) [58] are consistent with our results. Their analyses of old Italian apple cultivars revealed FRAP values ranging from 82 to 1727 µM/L. In contrast, the cultivars examined in our study exhibited notably higher antioxidant capacities, with the lowest recorded FRAP value being 193.35 µM/L and the highest reaching 3779.00 µM/L, far exceeding the values reported for Italian cultivars. It is important to highlight that the ‘Máté Dénes‘ cultivar demonstrated exceptional antioxidant potential even among the cultivars we analyzed.

### 2.3. Consumer Evaluations of Old, Local Apple Varieties

#### 2.3.1. Shape and Size

The shapes and sizes of old, local apple cultivars varied widely based on consumer evaluations, with some being small and round, while others were large and oblong.

Based on the opinions of the reviewers, only three of the old, local apple cultivars (‘Batul’, ‘Harang alma’, and ‘Daru sóvári’) attained points for the shapes and sizes of the fruits similar to those for the commercial cultivars (Figure 1). ‘Máté Dénes’ was awarded the lowest points: the shapes and sizes of its fruit were the least liked by the reviewers.

#### 2.3.2. Texture of the Flesh

The texture of the fruit of the old, local apple cultivars, according to consumer evaluations, ranged from crisp and firm to tender and mealy.

The texture of the flesh of the ‘Londoni pepin’, ‘Batul’, ‘Harang alma’, ‘Marosszéki piros’, ‘Galambka’, ‘Budai Ignác’, and ‘Bereczki Máté’ cultivars was found to be similar to that of ‘Golden Delicious’ based on consumer evaluation (Figure 2). The texture of the ‘Gyógyi piros’ and ‘Sóvári nobil’ fruit was the least liked by the reviewers.

#### 2.3.3. Skin Thickness

Based on the consumer evaluations, the skin thickness of the old, local apple cultivars varied, with some having thin and delicate skin, while others had thick and tough skin.

Upon comparing the skin thicknesses of the old, local apple cultivars, we found that most of the old cultivars reached the values of the commercial cultivars (Figure 3). The reviewers found the skin thickness of ‘Harang alma’ to be the best, while that of the ‘Máté Dénes’ cultivar was the worst.

#### 2.3.4. Dominant Skin Color

Based on consumer evaluations, the skin color of the old, local apple cultivars varied widely, with some being red or green, while others were yellow or even striped.

Based on the opinions of the reviewers, only three old, local apple cultivars (‘Simonffy Piros’, ‘Harang alma’, and ‘Daru sóvári’) received points for fruit skin color similar to those for the commercial cultivars (Figure 4). ‘Máté Dénes’ was awarded the lowest value; its skin color was the least liked by the reviewers.

#### 2.3.5. Taste of the Old, Local Apple Cultivars

Among the parameters evaluated, taste is one of the most significant for the customers. Much like Felföldi et al. [59], we found significant differences between the cultivars.

The taste of the old, local apple cultivars, as evaluated by the consumers, varied from tart and tangy to sweet and mellow, with each cultivar possessing a unique flavor profile.

In terms of taste, the ‘Marosszéki piros’ cultivar received the best score, which was similar to that of the ‘Watson Jonathan’ cultivar, while the ‘Máté Dénes’ cultivar received the worst score (Figure 5).

#### 2.3.6. Mean Rank

The Mean Rank for the old, local apple cultivars based on the consumer evaluations was useful in determining which cultivars were most preferred by the testers, providing valuable information for growers and customers. In terms of Mean Rank, the ‘Harang alma’ cultivar almost reached the ‘Watson Jonathan’ apple cultivar, while the reviewers liked ‘Máté Dénes’ the least (Figure 6).

Overall, we can conclude that the unique physicochemical properties of traditional varieties, along with their favorable customer perception, make them particularly appealing to specific niche markets. Moreover, the exceptional nutritional compositions of certain varieties render them highly suitable for health-conscious customers, supporting efforts toward healthier diets and sustainable food consumption.

Additionally, these cultivars are also known for their ability to tolerate environmental stressors and pathogens [7]: Papp et al. (2016) [60] found that ten apple cultivars obtained from the Carpathian Basin under field conditions (derived from their university’s ex situ germ plasm collection) had resistance to apple scab and powdery mildew. Furthermore, they examined the genes granting resistance to apple scab through a genetic analysis. According to their results, the apple genotypes examined had extraordinary resistance to both diseases [60,61].

## 3. Materials and Methods

### 3.1. The Cultivation Conditions for the Research Material

Apple samples were collected on the Experimental and Research Farm of the Department of Fruit Science of the Hungarian University of Agriculture and Life Sciences (Soroksár: N 47° 40′, E 19° 15′). The climatic conditions in the year of examination (2023) were dry and continental, with 550–600 mm of precipitation in an unbalanced distribution. The solar radiation during the summer was relatively high. An irrigation system had not been installed in the orchard. The soil type was sandy loam with high calcium content and a pH value of 7.7–7.8, determined using the KCl pH measurement method. However, the organic matter content was low, with a humus concentration of approximately 1%. The tested trees were planted in 2005, so they were in the 18th-leaf stage and still in good phytosanitary condition. The trees were grafted onto medium vigorous MM106 rootstocks and trained using the free-spindle training system, with a 4.5 m row distance by 4 m tree distance. In terms of growing technology, winter pruning was applied, but fruit thinning was not performed.

### 3.2. Plant Material

The fruits of eighteen old apple cultivars (‘Batul’, ‘Berecki Máté’, ‘Budai Ignác’, Daru sóvári’, ‘Fekete tányéralma’, ‘Galmabka’, ‘Gyógyi piros’, ‘Harang alma’ ‘Herceg Batthyányi’, ‘Kisasszony alma’, ‘Kiss Ernő tábornok’, ‘Londoni pepin’, ‘Marosszéki piros’, ‘Máté Dénes’, ‘Pusztai sárga’, ‘Simonffy Piros’, ‘Sóvári nobil’, and ‘Tükör alma’) were included in the trial. The optimal harvest time was the time suitable for storage (based on SSC and TTA as well as organoleptic examination). All the varieties examined required storage in a post-harvest facility. The standard for the trial was ‘Watson Jonathan’. The quality of the fruit of the ‘Watson Jonathan’ cultivar is still well-known in Central Europe, where it is popular. Central European customers (mainly Hungarians) prefer this fruit’s size and flavor characteristics, and this cultivar had greater importance in production in the past. Furthermore, the food industry is also seeking this cultivar because it is an excellent raw material for concentrate (it has high sugar and total acidity content). The ‘Watson Jonathan’ is not a global cultivar; it was dominant only in Central Europe for a few decades.

### 3.3. Physicochemical Parameters

For each cultivar, fifteen pieces of fruit were selected for analysis. The sampling time was determined based on continuous monitoring of fruit development, considering SSC, TA, flesh firmness, taste, and stem separation. The storage-appropriate ripeness characteristic of each cultivar was established based on these parameters. The fruit mass was determined using a digital scale (KPZ-2-05-4/6000, Klaus-Peter Zander GmbH, Hamburg, Germany). Flesh firmness was assessed using a Magness-Taylor hand penetrometer equipped with an 11 mm diameter probe. Prior to measurement, the skin on both the sun-exposed and shaded sides of the fruit was carefully removed using a blade. Firmness values were recorded in these areas, and an average was calculated.

The fresh fruits were homogenized using a blender and filtered through pleated filter paper (D = 18.5 mm; pore size: 7–10 µm; and thickness: 170 µm). The filtrate obtained through this process was subsequently analyzed in further examinations. Determination of soluble solids content was performed based on a [62] Codex Alimetarius 3-1-558/93 formula in °Brix (g 100 g^−1^) with a digital refractometer (ATAGO Palette PR-10, Atago Co., Ltd., Tokyo, Japan), using repetitions to ensure accuracy. Total titratable acid content was determined following the Nr. 12147:1998 [63] Hungarian Standard, given in malic acid equivalents (m/m%), based on six repeated measurements. The sugar/acid ratio was calculated from the total soluble solids and the titratable acid content.

### 3.4. Biologically Active Compounds

The samples (15 fruits/cultivars) were homogenized in a blender and stored in a freezer at −28 °C until analysis. After removal from the freezer, the biologically active compound content of the samples was measured in three repetitions.

The polyphenol content was measured with Folin–Ciocalteu reagents at λ = 765 nm using spectrophotometer (Hitachi UV/VIS U-2800 A spectrophotometer (Tokyo, Japan)). The calibration curve was made based on gallic acid using a method provided in [64]; three replicates were employed each time.

The FRAP (Ferric Reducing Ability of Plasma) assay was carried out according to the method noted in [65]. The FRAP assay is based on the reduction of the Fe^3+^-2,4,6-tripyridyl-S-triazine complex to the ferrous form (Fe^2+^), and the intensity of the reaction is monitored by measuring the change in absorption at 593 nm.

The pectin content was determined using the method described in [66] at 525 nm using Hitachi U.2800 spectrophotometer.

### 3.5. Consumer Evaluation

Consumer evaluations of the fruits were carried out in collaboration with young people in their 20s, namely, 14 men and 12 women, all non-professional reviewers, according to the procedure described by Vindras et al. [67]. The samples were examined post-ripening (at 2 ± 0.2 °C and 85% humidity under normal atmospheric pressure) and after three months in a normal-atmosphere cold storage unit until reaching maturity. The washed fruits (ten pieces) were presented whole, including the skin. Slicing was carried out by the evaluators themselves. The traits examined were the following: each fruit’s dominant color, size, shape, taste, and flavor, as well as the color of the peel, skin thickness, and texture of the flesh [59,67,68]. The participants had to evaluate various aspects of the fruit on a hedonic evaluation scale from 1 to 5, with 1 being the lowest rating and 5 being the highest rating. Definitions of the examined traits were determined by the reviewers (costumers who support the fruit business financially) based on their subjective opinions; however, they were given a brief explanation of how the scores are awarded. To minimize any potential biases during assessment, we provided no specific guidelines, allowing participants to make their decisions based on their own impressions and preferences. Based on the research by Chambers and Koppel (2013) [69], a general guideline can be consulted to explain the specific attributes.

### 3.6. Statistical Methods

The statistical analysis was conducted using IBM SPSS Statistics 20. Univariate ANOVA was conducted to separate the homogenous groups using the Tukey test. The RSD (relative standard deviation) value of the determination was 5%. Cobweb diagrams were prepared based on each of the properties evaluated. All the scores assigned to the cultivars were evaluated through non-parameterized tests based on rank numbers using the Friedman test for K-related samples, on the basis of which a ranking was established.

## 4. Conclusions

Based on our results, the unique physicochemical properties of traditional varieties, along with their favorable perception by customers, make them particularly appealing to specific niche markets. Moreover, the exceptional nutritional compositions of certain varieties render them highly suitable for health-conscious customers, supporting efforts toward healthier diets and sustainable food consumption.

Additionally, these cultivars are also known for their ability to tolerate environmental stressors and pathogens [7]: Papp et al. (2016) [60] found that ten apple cultivars obtained from the Carpathian Basin under field conditions (derived from their university’s ex situ germ plasm collection) had resistance to apple scab and powdery mildew. Furthermore, they examined the genes granting resistance to apple scab through a genetic analysis. According to their results, the examined apple genotypes had extraordinary resistance to both diseases [60].

Based on the results of previous research, the ‘Batul’, ‘Vilmos renet’, ‘Pónyik’, ‘Sikulai’, ‘Tordai piros kálvil’, and ‘Szabadkai szercsika’ cultivars are recommended for ecological production because of their excellent disease resistance and suitable fruit quality [5,7,60,70].

This research confirms that the ‘Batul’ cultivar has desirable traits for customers, can serve as a gene source for breeding programs, and contains outstanding biological active compounds.

‘Máté Dénes‘ was found to contain extremely large quantities of biologically active ingredients, but the customer preference for this variety was poor. But it may become an important gene source of breeding programs based on its biologically active compounds.

Based on the excellent fruit quality and customer preferences observed, we can recommend ‘Harang alma’ and ‘Marosszéki piros’ for organic farming or backyard gardens.

Although ’Pusztai sárga’ did not show excellent fruit quality with respect to either of the parameters, its quality was similar to that of the commercial cultivars. Furthermore, as our earlier research shows, ‘Pusztai sárga’ can tolerate fire blight. That is why we can recommend it for use in organic farming.

The large degree of trait variability ensures these cultivars’ suitability as basic material for breeding [5]. Local apple cultivars have developed tolerance to environmental stress factors, making their re-evaluation crucial for sustaining apple production in the Carpathian Basin [11,60,70,71,72].

Importantly, preserving and cultivating diverse, locally adapted apple varieties can contribute to sustainable agriculture and help address the challenges posed by climate change by providing genetic resources for apple breeding.

## Figures and Tables

**Figure 1 plants-14-02005-f001:**
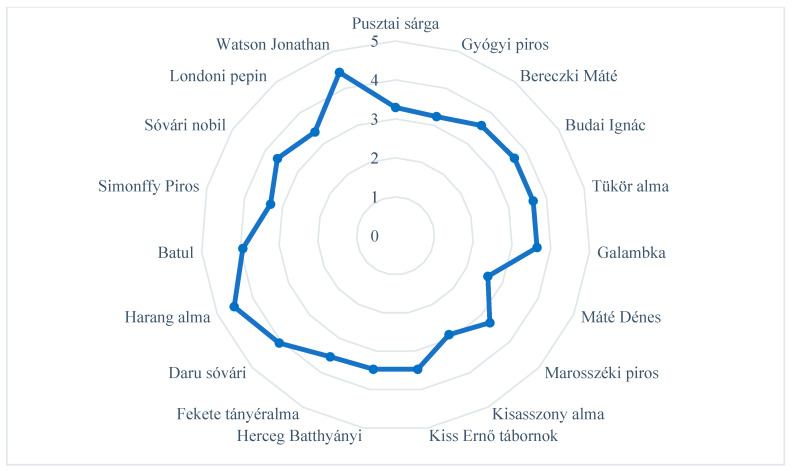
Appearance (shapes and sizes) of old, local apple cultivars evaluated in experimental trials based on consumer evaluation. Statistical differences were assessed using the Friedman test. Significance level: *p* < 0.05.

**Figure 2 plants-14-02005-f002:**
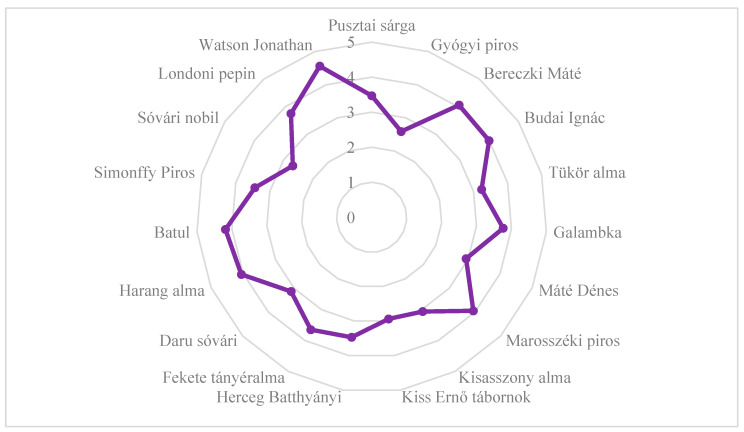
Texture of the fruit of old, local apple cultivars evaluated in experimental trials based on consumer evaluations. Statistical differences were assessed using the Friedman test. Significance level: *p* < 0.05.

**Figure 3 plants-14-02005-f003:**
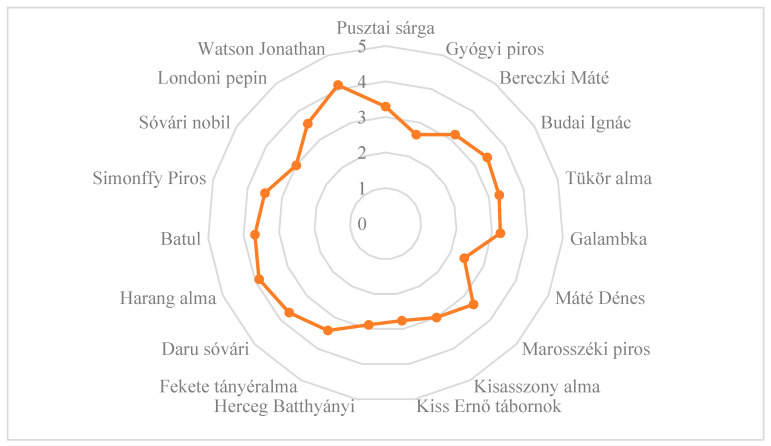
The skin thickness of old, local apple cultivars evaluated in experimental trials based on consumer evaluations. Statistical differences were assessed using the Friedman test. Significance level: *p* < 0.05.

**Figure 4 plants-14-02005-f004:**
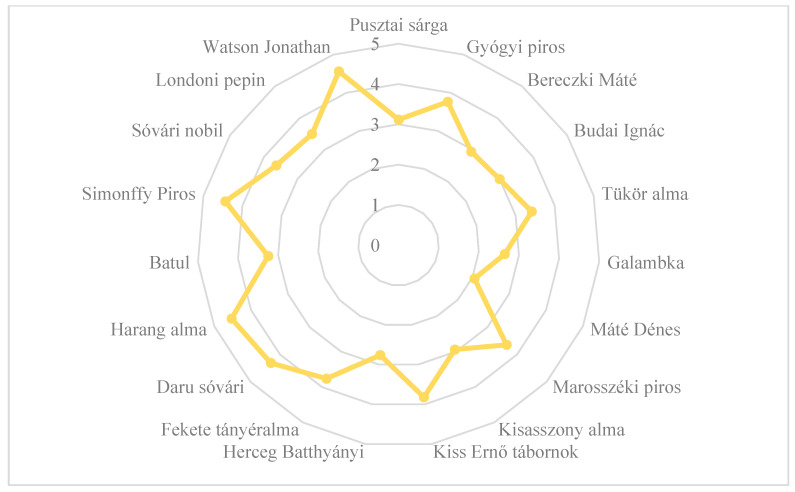
The dominant skin color of old, local apple cultivars evaluated in experimental trials based on consumer evaluation. Statistical differences were assessed using the Friedman test. Significance level: *p* < 0.05.

**Figure 5 plants-14-02005-f005:**
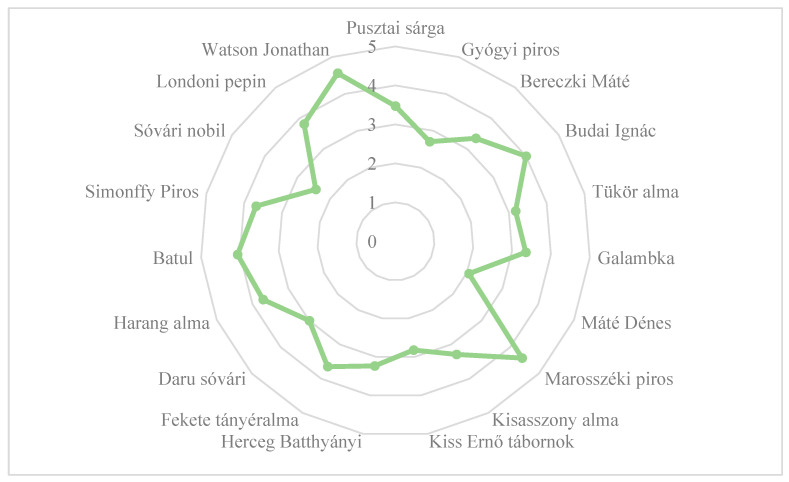
The taste of old, local apple cultivars evaluated by consumers in experimental trials. Statistical differences were assessed using the Friedman test. Significance level: *p* < 0.05.

**Figure 6 plants-14-02005-f006:**
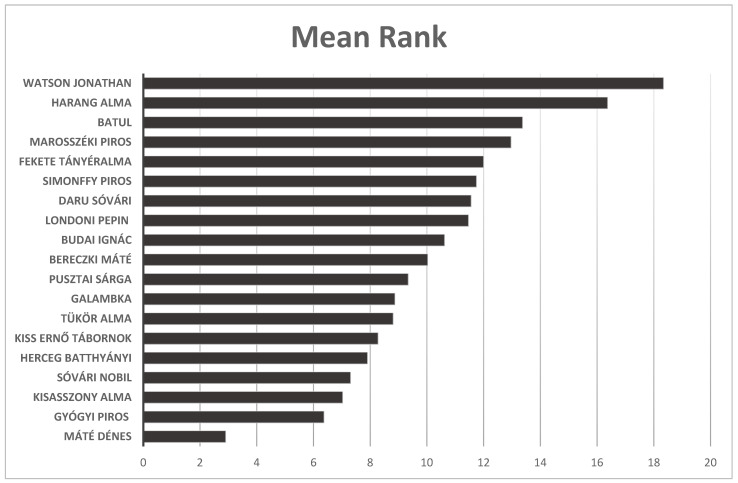
The Mean Rank for the old, local apple cultivars evaluated by consumers in experimental trials. Statistical differences were assessed using the Friedman test. Significance level: *p* < 0.05.

**Table 1 plants-14-02005-t001:** The physicochemical parameters of the old apple cultivars evaluated in experimental trials.

Cultivar	Fruit Mass (g)	Firmness (kg cm^−2^)	SSC (°Brix)	TA (%)	SSC/TA
Batul	175 ± 15.45 de	7.5 ± 0.9 b–e	14.5 ± 0.3 c–g	0.72 ± 0.02 e–g	20.2 ± 0.35 d
Bereczki Máté	163 ± 13.48 d	9.9 ± 1.0 f–h	14.1 ± 0.4 b–f	0.77 ± 0.03 g	18.4 ± 1.49 cd
Budai Ignác	140 ± 8.32 bc	8.5 ± 1.0 d–f	15.2 ± 0.2 e–h	0.88 ± 0.01 h	17.3 ± 1.91 c
Daru sóvári	155 ± 10.12 c	6.2 ± 1.4 ac	13.6 ± 0.2 a–e	0.45 ± 0.41 bc	30.5 ± 0.72 f
Fekete tányéralma	157 ± 9.11 c	10.7 ± 0.9 gh	12.9 ± 0.1 ab	0.67 ± 0.03 e–g	19.3 ± 1.06 d
Galambka	137 ± 9.38 b	6.1 ± 0.3 ac	15.1 ± 0.2 d–h	0.45 ± 0.04 bc	33.4 ± 3.4 fg
Gyógyi piros	128 ± 6.53 ab	5.8 ± 0.7 ab	13.4 ± 0.3 a–d	0.90 ± 0.01 h	14.9 ± 0.29 a
Harang alma	143 ± 8.11 bc	6.5 ± 1.1 a–c	14.2 ± 0.1 b–g	0.44 ± 0.02 b	32.0 ± 0.94 fg
Herceg Battyán	120 ± 7.32 ab	8.8 ± 1.0 e–g	16.9 ± 0.3 ij	1.09 ± 1.1 i	15.4 ± 0.61 ab
Kis Ernő tábornok	142 ± 11.22 bc	11.4 ± 0.6 h	15.7 ± 1.3 g–i	0.74 ± 0.08 g	21.1 ± 2.09 de
Kisasszony alma	133 ± 15.84 b	6.4 ± 1.3 ac	14.4 ± 0.2 b–g	0.62 ± 0.01 de	23.2 ± 0.57 ef
Londoni pepin	145 ± 7.49 bc	7.5 ± 0.9 b–e	15.1 ± 0.1 e–h	0.63 ± 0.03 d–f	23.9 ± 1.44 ef
Marosszéki piros	168 ± 14.52 cd	8.9 ± 0.7 e–g	14.2 ± 0.2 b–g	0.90 ± 0.02 h	15.8 ± 0.57 bc
Máté Dénes	153 ± 12.86 c	5.5 ± 0.9 a	17.4 ± 0.4 j	0.22 ± 0.02 a	80.7 ± 8.04 h
Pusztai sárga	146 ± 7.19 bc	11.2 ± 0.8 h	15.6 ± 0.3 f–i	0.73 ± 0.02 fg	21.4 ± 0.70 de
Simonffi piros	105 ± 5.13 a	6.7 ± 0.9 a–c	13.3 ± 0.4 a–c	0.69 ± 0.02 e–g	19.1 ± 0.09 d
Sóvári nobil	128 ± 11.24 ab	6.7 ± 1.1 a–d	14.1 ± 0.2 b–g	0.55 ± 0.01 cd	25.8 ± 0.79 ef
Tükör alma	157 ± 12.69 c	7.6 ± 1.1 b–e	12.4 ± 0.2 a	0.68 ± 0.02 e–g	18.4 ± 0.51 cd
Watson Jonathan	140 ± 5.78 bc	8.1 ± 0.6 c–f	16.4 ± 0.6 h–j	0.90 ± 0.03 h	18.1 ± 0.92 cd

‘Watson Jonathan’ was the control cultivar, which we have highlighted using a grey background color in the table. Different letters indicate significant differences (Tukey’s, *p* < 0.05, a–j homogeneity groups).

**Table 2 plants-14-02005-t002:** The health-promoting properties of the old, local apple cultivars evaluated in the experimental trials.

Cultivars	Pectin (%)	Polyphenol (mg GAE/l)	FRAP (µM/L)
Batul	0.73 ± 0.04 a–d	673.03 ± 14.99 i	650.84 ± 5.93 cd
Bereczky Máté	0.35 ± 0.02 a	181.92 ± 10.63 a	226.31 ± 3.29 a
Budai Ignác	0.92 ± 0.21 c–f	468.83 ± 36.86 f	288.25 ± 12.52 a
Daru Sóvári	0.53 ± 0.03 a–c	339.67 ± 16.27 cd	283.64 ± 1.32 a
Fekete tányér alma	0.61 ± 0.11 a–d	293.04 ± 32.79 b	273.75 ± 4.61 a
Galambka	1.26 ± 0.01 f	877.17 ± 57.25 j	1606.74 ± 159.15 g
Gyógyi Piros	0.59 ± 0.14 a–d	362.17 ± 18.43 de	525.23 ± 0.98 bc
Harang alma	0.98 ± 0.03 d–f	398.83 ± 16.27 e	536.17 ± 35.59 bc
Herceg Battyán	0.57 ± 0.10 a–c	721.33 ± 13.78 i	776.38 ± 72.16 de
Kis Ernő Tábornok	0.61 ± 0.16 a–d	451.33 ± 44.04 f	865.53 ± 0.02 b
Kisasszony alma	0.55 ± 0.03 a–c	503.03 ± 54.08 fg	865.35 ± 50.41 ef
Londoni Pepin	0.45 ± 0.001 ab	696.33 ± 42.89 i	906.86 ± 8.89 ef
Marosszéki Piros	1.19 ± 0.43 e–f	290.67 ± 29.43 c	193.35 ± 8.56 a
Máté Dénes	0.80 ± 0.01 b–e	1956.33 ± 26.96 k	3779.00 ± 318.30 h
Pusztai Sárga	0.68 ± 0.05 a–d	313.01 ± 30.41 b–d	332.54 ± 3.95 a
Simonffi Piros	0.60 ± 0.09 a–d	555.51 ± 15.21 gh	672.59 ± 19.77 cd
Sóvári nobil	0.45 ± 0.05 ab	584.67 ± 28.76 h	942.45 ± 30.64 f
Tükör Alma	0.37 ± 0.05 a	534.67 ± 16.65 gh	697.30 ± 22.74 d
Watson Jonathan	0.59 ± 0.05 a–c	464.42 ± 27.17 f	764.72 ± 10.54 de

‘Watson Jonathan’ was the control cultivar, which we have highlighted using a grey background color in the table. Different letters indicate significant differences (Tukey’s, *p* < 0.05, a–k homogeneity groups).

## Data Availability

Data are contained within the article.

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
