# Peer review of "Assessing the Post-Brief-Storage Fruit Quality and Sensory Characteristics of Old, Local Apple Cultivars from the Carpathian Basin"

_plants, 2025, doi:10.3390/plants14132005_

Round 1
Reviewer 1 Report
Comments and Suggestions for Authors
Page 1
Title:
Assessing fruit… …storage period, and… …apple cultivars…
There is no assessment of climate change so the title is misleading
Abstract:
Reflecting the norm in global apple…
There is no summary of climate change in the abstract
Introduction:
Delete based on numerous pomological studies
Old apple cultivars…
Page 2
Delete only some
Old apple cultivars have…
Genetic diversity not biodiversity provides resistance to disease
…to increase the genetic diversity…
First sentence of the second paragraph doesn’t make sence.
germplasm
save
genetic diversity
…collect accessions[14]…
Delete the multiple use of the word ‘in’ before each country
And Irand
…find, collect, preserve?, maintain…
Have been demonstrated by several studie…
Delete ‘as a result of which we can observe.’
Not sure how the fourth paragraph on page two links to the third paragraph. Is an abrupt topic change.
Page 3
…native apple cultivars have adapted to…
…development of distinct genotypes…
Popular instead of well-preferred ?
Delete ‘final’
Unsure of journal policy, do you need the author citations for scientific names?
What is meant by ‘food safety’
Final paragraph of the introduction is informally written, suggest re-write
Remove italics from Physicochemical
The heading for table one lacks all the detail required. You should be able to understand where and what the data is by reading the heading of the table alone.
Page 4
I question the ability to discuss fruit size differences as there is no evidence of cropload being controlled and therefore the data is likely confounded
As all the trees were trained to the same system and in the same orchard is it reasonable to expect major environmental factor differences?
Fruit firmness is an important characteristic of…
Is firmness assessed at harvest?
…point of view of consumers…
Provide a citation for the 6-7 kg/cm2 figure
Rather than referring to Watson Jonathan as a control, perhaps standard would be better
What is meant by ‘minimum border’?
Final sentence of second paragraph – would like you to explain this in more detail as firmness is a key trait for consumers.
Don’t refer to Brix values, us your abbreviation SSC. You would never say ‘ Grams values of the apple cultivars varied between 12 and 17 g.
The comment regarding Mate Denes should be in the sugar/acid paragraph
Don’t need ‘(°Brix)’
Was cropload controlled in the polish study? If not is this a reliable source
Page 5
…than our data.
…variability in SSC values.
The greatest acid content was measured in Herceg Battyan and the least in Mate Denes.
What are the important differences?
Instead of the term flavour, use perceived sweetness
Table two heading lacks details
Page 7
I have no understanding of the scale or units in Figures 1 to 6. It is also unclear if each cultivar has its own descriptors for each trait. For example, are they all red apples, or do they have different colour. If they are different how does the scale in Figure 4 work?
How can you be showing both shape and size in Fig 1?
I think bar charts would be better for Figs 1-4
None of the figure captions have sufficient detail
Page 11
Don’t understand this sentence. ‘ typical for all oth them, that all check varieties required post-harvest facility.’
Rather than control, suggest use of the term standard
Were there replicate trees or each cultivar, if not, you have pseudo-replication
When was the fruit harvested. Were all cultivars picked at the same time or were they maturity picked? What criteria was used? Perhaps your differences could be due to harvest date? Back ground colour and or starch pattern index would help considerably.
Page 12
You should really provide the details of the scales used.
…reviewers (final consumers)…
Page 13
Conclusions
…grown apple cultivars, originating from the Carpathian Basin, exhibit favourable….
…high potential for yielding… - you haven’t calculated yield, nor have you controlled cropload. I disagree with this statement.
Page 14
…Our research confirmed, that Batul ahd desirable consumer traits and …
…gene source for breeding programs…
Do you have breeding values to support this statement?
I don’t follow the recommendation for organic farming, as none of your data relates to this.
Provide a citation for earlier research
Second to last paragraph
You have not shown genetic variability, but rather trait variability.
Comments on the Quality of English LanguageSome minor improvments to English required, as per comments above
Author Response
Dear Reviewer1,
the authors are very grateful for your time, suggestion, and all of your comments, which you provided us to improve the quality of the manuscript, reviewed by you. We appreciate all of your activities, and almost all of your comments were accepted.
Here you can find the detailed answers by pages.
Comment to page 1
Page 1
Title:
Assessing fruit… …storage period, and… …apple cultivars…
There is no assessment of climate change so the title is misleading
Abstract:
Reflecting the norm in global apple…
There is no summary of climate change in the abstract
Introduction:
Delete based on numerous pomological studies
Old apple cultivars…
Answers to reviewer’s comments to page 1
The authors revised the requested words, and sentences. We deleted the climate change from the summary.
Comments to Page 2
Delete only some
Old apple cultivars have…
Genetic diversity not biodiversity provides resistance to disease
…to increase the genetic diversity…
First sentence of the second paragraph doesn’t make sence.
germplasm
save
genetic diversity
…collect accessions[14]…
Delete the multiple use of the word ‘in’ before each country
And Irand
…find, collect, preserve?, maintain…
Have been demonstrated by several studie…
Delete ‘as a result of which we can observe.’
Not sure how the fourth paragraph on page two links to the third paragraph. Is an abrupt topic change.
Answers to reviewer’s comments to page 2
The authors revised the requested sentences, and we accepted all of your suggestions. We added a paragraph (fourth paragraph on this page) about the control. It was important to mention its importance, because it is a unique one. In countries, located in the Carpathian Basin, this cultivar has good remoné and importance, and it is typical for this part of Europe.
Comments to Page 3
…native apple cultivars have adapted to…
…development of distinct genotypes…
Popular instead of well-preferred ?
Delete ‘final’
Unsure of journal policy, do you need the author citations for scientific names?
What is meant by ‘food safety’
Final paragraph of the introduction is informally written, suggest re-write
Remove italics from Physicochemical
The heading for table one lacks all the detail required. You should be able to understand where and what the data is by reading the heading of the table alone.
Answers to reviewer’s comments to page 3
We accepted your comments, and suggestions. We corrected the table, can be found on this page. We deleted the sentences containing your first two comments on this section.
Comments to Page 4
I question the ability to discuss fruit size differences as there is no evidence of cropload being controlled and therefore the data is likely confounded
As all the trees were trained to the same system and in the same orchard is it reasonable to expect major environmental factor differences?
Fruit firmness is an important characteristic of…
Is firmness assessed at harvest?
…point of view of consumers…
Provide a citation for the 6-7 kg/cm2 figure
Rather than referring to Watson Jonathan as a control, perhaps standard would be better
What is meant by ‘minimum border’?
Final sentence of second paragraph – would like you to explain this in more detail as firmness is a key trait for consumers.
Don’t refer to Brix values, us your abbreviation SSC. You would never say ‘ Grams values of the apple cultivars varied between 12 and 17 g.
The comment regarding Mate Denes should be in the sugar/acid paragraph
Don’t need ‘(°Brix)’
Was cropload controlled in the polish study? If not is this a reliable source
Answers to reviewer’s comments to page 4
All of your comments were accepted, and we revised this page. Many Thanks to suggest us more ways to avoid using the same words.
Yes, you are right, the cropload have influence to the fruit size. In our trial, there was not any regulation on the cropload, so the examined cultivars could show their natural fruit size.
The ‘minimum border’ was revised. Meaning of this is the lowest value, which can be accepted.
There was no data about the cropload control in the Polish study.
Comments to Page 5
…than our data.
…variability in SSC values.
The greatest acid content was measured in Herceg Battyan and the least in Mate Denes.
What are the important differences?
Instead of the term flavour, use perceived sweetness
Table two heading lacks details
Answers to reviewer’s comments to page5
Everything was accepted on this page.
Comments to Page 7
I have no understanding of the scale or units in Figures 1 to 6. It is also unclear if each cultivar has its own descriptors for each trait. For example, are they all red apples, or do they have different colour. If they are different how does the scale in Figure 4 work?
How can you be showing both shape and size in Fig 1?
I think bar charts would be better for Figs 1-4
None of the figure captions have sufficient detail
Answers to reviewer’s comments to page7
The values, during the consumer test, were ranged from 1 to 5. In other, later published research, this type of diagram was used to present the results, therefore the authors decided to choose.
The shape and size meant appearance. We added some more data to the figures.
Comments to Page 11
Don’t understand this sentence. ‘ typical for all oth them, that all check varieties required post-harvest facility.’
Rather than control, suggest use of the term standard
Were there replicate trees or each cultivar, if not, you have pseudo-replication
When was the fruit harvested. Were all cultivars picked at the same time or were they maturity picked? What criteria was used? Perhaps your differences could be due to harvest date? Back ground colour and or starch pattern index would help considerably.
Answers to reviewer’s comments to page11
All examined cultivars required post-harvest facility.
Yes, there were replicate trees.
The cultivars were picked not at the same time, but we harvested them on their harvest maturity.
Answers to Page 12
You should really provide the details of the scales used.
…reviewers (final consumers)…
Comments to Page12
The categories were given to the reviewers, and we used consumer evaluation test. So the reviewers received some basic guidelines about the test. We decided to evaluate the samples from 1 to 5 by their own impressions preferences.
Answers to Page 13
Conclusions
…grown apple cultivars, originating from the Carpathian Basin, exhibit favourable….
…high potential for yielding… - you haven’t calculated yield, nor have you controlled cropload. I disagree with this statement.
Comments to Page13
We deleted these two sentences form the text.
Answers to Page 14
…Our research confirmed, that Batul ahd desirable consumer traits and …
…gene source for breeding programs…
Do you have breeding values to support this statement?
I don’t follow the recommendation for organic farming, as none of your data relates to this.
Provide a citation for earlier research
Second to last paragraph
You have not shown genetic variability, but rather trait variability.
Comments to Page14
We accepted all of your comments.
We hope that our answers match to your expectations.
Yours sincerely,
the authors

Reviewer 2 Report
Comments and Suggestions for Authors
- line 80, what's after 'as a result of which we can observe'? Is this paragraph over?
- In Table 1, the letters' abcd 'that describe significance should be close to the numerical annotations. Also, try not to place the table across pages. Why is a background color added to the last row of the table?
- Line 161, is' Sugar/acid 'a title? If so, please add a separate title.
- Line 174, the problem of table 2 is the same as that of table 1. Besides, why is there an additional horizontal line in the last row of Table 2?
- The figures 1 to 5 are not aesthetically pleasing. It is suggested that they be beautified. The font should be times new roman as much as possible.
- Could the format of references be standardized? Should the names of journals be abbreviated or in full pinyin? Some references do not have a doi.
- Line 33, there should be no punctuation marks after the last keyword.
- Line 60. I think this paragraph and the next paragraph should not talk about the background of various places at length. The word count should be appropriately reduced.
- In Table 1, the author determined the physicochemical parameters of some old varieties, but the differences from the world-famous varieties such as' Golden Delicious', 'Red Delicious',' Idared 'and' Gala 'were not reflected here. Is it necessary to use these varieties as controls to measure the old varieties in the article?
- Line 185, the indentation format of the first line of the paragraph does not match that of other paragraphs.
- For Line 366, is the writing of 'Fe3+' and 'Fe2+' correct?
- There are problems with the font format and paragraph spacing from line 390 to line 400. Please unify the entire text.
- The fourth part of the article should not merely be the conclusion. The author has also conducted some discussions in it. The title should be changed to Discussion and Conclusion. Or the conclusion can be presented as a separate paragraph, providing a simple summary.
- It is suggested that the author polish the language of the article.

Author Response
Dear Reviwer2,
first of all, authors of this manuscript appreciate all of your time, suggestions, and comments, which you provided in your review. We believe, that all of your activities helped us a lot to increase quality of the paper.
Here you can find our answer to your comments
Comment1. line 80, what's after 'as a result of which we can observe'? Is this paragraph over?
Answer to the comment1: Yes, it is.
Comment2: In Table 1, the letters' abcd 'that describe significance should be close to the numerical annotations. Also, try not to place the table across pages. Why is a background color added to the last row of the table?
Answer to the comment2: The background color was a mistake, made by us. In the revised version, we deleted the background color. Many Thanks to call our attention for it.
Comment3: Line 161, is' Sugar/acid 'a title? If so, please add a separate title.
Answer to the comment3: It was deleted, because it is not a separate title.
Comment4: Line 174, the problem of table 2 is the same as that of table 1. Besides, why is there an additional horizontal line in the last row of Table 2?
Answer to the comment4: The additional horizontal line marks the control.
Comment5: The figures 1 to 5 are not aesthetically pleasing. It is suggested that they be beautified. The font should be times new roman as much as possible.
Answer to the comment5: Thank you for your comment, it was accepted.
Comment6: Could the format of references be standardized? Should the names of journals be abbreviated or in full pinyin? Some references do not have a doi.
Answer to the comment6: The authors used Web of Science Help to find the abbreviatied names of the journals. Some journals don’t have abbreviated names, therefore we put its original ones. The situation is the same with doi-s.
Comment7: Line 33, there should be no punctuation marks after the last keyword.
Answer to the comment7: It was accepted.
Comment8: Line 60. I think this paragraph and the next paragraph should not talk about the background of various places at length. The word count should be appropriately reduced.
Answer to the comment8: It was accepted.
Comment 9: In Table 1, the author determined the physicochemical parameters of some old varieties, but the differences from the world-famous varieties such as' Golden Delicious', 'Red Delicious',' Idared 'and' Gala 'were not reflected here. Is it necessary to use these varieties as controls to measure the old varieties in the article?
Answer to the comment9: The text was revised.
Comment10: Line 185, the indentation format of the first line of the paragraph does not match that of other paragraphs.
Answer to the comment10: It was accepted.
Comment11: For Line 366, is the writing of 'Fe3+' and 'Fe2+' correct?
Answer to the comment11: It was corrected.
Comment12: There are problems with the font format and paragraph spacing from line 390 to line 400. Please unify the entire text.
Answer to the comment2: It was revised.
Comment13: The fourth part of the article should not merely be the conclusion. The author has also conducted some discussions in it. The title should be changed to Discussion and Conclusion. Or the conclusion can be presented as a separate paragraph, providing a simple summary.
Answer to the comment13: It was revised.
Comment14: It is suggested that the author polish the language of the article.
Answer to the comment14: The text was revised by a native speaker.
We hope that our answers match to your expectations.
Yours sincerely,
the authors

Reviewer 3 Report
Comments and Suggestions for Authors
Dear Authors,
The submitted manuscript entitled "Assessing the Fruit Quality after Short-Term Storage and Sensory Characteristics of Old, Local Apple Varieties Derived from the Carpathian Basin in Response to Climate Change" covers an interesting and timely topic.
The Introduction briefly prepares the reader for the subsequent sections. The authors present statistically analyzed results, which are well described, discussed, and accompanied by broad conclusions. However, some significant issues need to be addressed before the manuscript can proceed to the next stage of the publication process.
The major limitations of this study are the small number of fruits used for evaluation and the low number of participants in the "sensory" analysis, which appears to be more of a consumer test than a formal sensory evaluation. Furthermore, a detailed revision reveals that the manuscript requires substantial improvement in both content and presentation. Taking this into account I recommend a major revision of the manuscript
Below are listed the most important issues:
- The term "weight" is used throughout the manuscript to describe a parameter that is more accurately referred to as "fruit mass." This terminology should be reviewed and corrected consistently across the entire manuscript.
- While the results are well presented in clear tables, the table titles and descriptions require improvement. For instance, “Table 1. The physicochemical parameters of the old apple cultivars” should be expanded to include “tested in the trial” for greater specificity. Additionally, footnotes should be added to explain the meaning of the letters used to indicate homogeneous groups, as is standard in scientific research papers.
- Line 194 – can it be said that apples are rich in FRAP? In my opinion apples can show high) (for example) FRAP, which still is rather common than scientific language. This should be imrpved.
- Line 287-316 – This subsection should be divided or other title should be used while not only plant materian is described here.
- Line 310 – The reported pH value appears relatively high for apples. It is unclear whether the pH was measured in KCl, distilled water, or using another solution. This detail is essential, as different methods can yield significantly different pH values and should therefore be explicitly stated.
- The sensory evaluation should ideally be conducted by trained, experienced panelists. Based on the description provided (especially in lines 330–340), the procedure followed resembles a consumer test rather than a proper sensory analysis. This distinction should be clearly acknowledged and addressed in the revised manuscript.
- Line 325 – More detailed information about the controlled atmosphere (CA) storage conditions is needed, particularly the specific CO₂:O₂ concentrations.
- Line 345 – Additional details are needed regarding the harvest date, decision criteria for harvest timing, the number of apples evaluated, and the methodology used for measuring fruit firmness. For example: What diameter of penetrometer tip was used? Were apples peeled prior to testing? How many replicates were performed per treatment? Were SSC and TA measured directly in fruit juice, or by another method? These methodological details should be added comprehensively to the Materials and Methods section.
- Line 348 – does Brix degree strictly means g/100g? As I am not sure I would be happy to get this information as a reply to this review.
Good luck!
Regards
Comments on the Quality of English LanguageAlthough I’ve noticed that the quality of English can sometimes be improved, I don’t feel qualified to comment on it in more detail.
Author Response
Dear Reviwer3,
first of all, authors of this manuscript appreciate all of your time, suggestions, and comments, which you provided in your review. We believe, that all of your activities helped us a lot to increase quality of the paper.
Here you can find our answer to your comments
Comment1: The term "weight" is used throughout the manuscript to describe a parameter that is more accurately referred to as "fruit mass." This terminology should be reviewed and corrected consistently across the entire manuscript.
Answer1: It was accepted.
Comment2: While the results are well presented in clear tables, the table titles and descriptions require improvement. For instance, “Table 1. The physicochemical parameters of the old apple cultivars” should be expanded to include “tested in the trial” for greater specificity. Additionally, footnotes should be added to explain the meaning of the letters used to indicate homogeneous groups, as is standard in scientific research papers.
Answer2: It was accepted.
Comment3: Line 194 – can it be said that apples are rich in FRAP? In my opinion apples can show high) (for example) FRAP, which still is rather common than scientific language. This should be imrpved.
Answer3: It was accepted.
Comment4: Line 287-316 – This subsection should be divided or other title should be used while not only plant materian is described here.
Answer4: It was accepted.
Comment5: Line 310 – The reported pH value appears relatively high for apples. It is unclear whether the pH was measured in KCl, distilled water, or using another solution. This detail is essential, as different methods can yield significantly different pH values and should therefore be explicitly stated.
Answer5: It was accepted.
Comment6: The sensory evaluation should ideally be conducted by trained, experienced panelists. Based on the description provided (especially in lines 330–340), the procedure followed resembles a consumer test rather than a proper sensory analysis. This distinction should be clearly acknowledged and addressed in the revised manuscript.
Answer6: It was corrected.
Comment7: Line 325 – More detailed information about the controlled atmosphere (CA) storage conditions is needed, particularly the specific CO₂:O₂ concentrations.
Answer7: It was extended.
Comment8: Line 345 – Additional details are needed regarding the harvest date, decision criteria for harvest timing, the number of apples evaluated, and the methodology used for measuring fruit firmness. For example: What diameter of penetrometer tip was used? Were apples peeled prior to testing? How many replicates were performed per treatment? Were SSC and TA measured directly in fruit juice, or by another method? These methodological details should be added comprehensively to the Materials and Methods section.
Answer8: It was accepted and extended.
Comment9: Line 348 – does Brix degree strictly means g/100g? As I am not sure I would be happy to get this information as a reply to this review.
Answer9: It was accepted.
We hope that our answers match to your expectations.
Yours sincerely,
the authors

Reviewer 4 Report
Comments and Suggestions for Authors
The experiment compares the most frequently assessed features of a dozen or so local apple varieties in Hungary. The authors emphasize that these features may change under the influence of environmental factors, i.e. years of research. Why did they not perform this analysis within two consecutive years? In the abstract, state which features specifically determined the distinction of these two varieties. In the methodology, they did not provide the number of repetitions for some biochemical parameters. The discussion can be extended to include other authors, this is a commonly studied issue. Provide ranges of results obtained by other authors. In the summary, describe only your own results.

Author Response
Dear Reviewer4,
authors of this manuscript appreciate all of your time, work, suggestions, and comments. We accepted all, and revised the manuscript.
Yours sincerely,
the authors
Round 2
Reviewer 3 Report
Comments and Suggestions for Authors
Dear Authors,
thank you for thwe work you have done so far.
Below you will find some more tips how to improve your manuscript.
Good luck!
Line 30 – there is „height pectin content” and should be ”high pectin content”
Line 114 – there is “the ‘Jonathan’ apple cultivar, and should be “the ‘Jonathan’ apple cultivar and its mutants” –while you used not ‘Jonathan’ but ‘Watson Jonathan’ in your research. To be honest I am not sure if it is still so popular in Central Europe. Give some reference here or change the sentence to underline its unique traits
Line 173 – there is “Podosphera leucotricha”and should be “Podosphaera leucotricha”
Line 254 – results presented in reference 32 you cited are not only Polish cultivars. Authors of the cited paper examined several old cultivars GROWN in Poland, not Polish ones
Table 2 – are you sure that pectin and FRAP can be defined as biologically active compounds? Please check it carefully.
Line 521,534 – I am not sure if apples can contain amount of, or can be a source of FRAP, please check it carefully
Line 845-849 – please give more precise information on method that were used to determine optimal harvest time. Your description is too general. Please also describe methods of assessing each parameter in the same order as you present it in the Results section.
Comments on the Quality of English LanguageI do not feel qualified to asess the quality of English language.
Author Response
Dear Reviewer3,
Many Thanks for your second review. The authors appreciate your time, all of your efforts, and suggestion to imporve the quality of this paper.
Here you can find our answers:
Line 30 – there is „height pectin content” and should be ”high pectin content”
Answer: Many Thanks for the correction, the correction have been made in the main text.
Line 114 – there is “the ‘Jonathan’ apple cultivar, and should be “the ‘Jonathan’ apple cultivar and its mutants” –while you used not ‘Jonathan’ but ‘Watson Jonathan’ in your research. To be honest I am not sure if it is still so popular in Central Europe. Give some reference here or change the sentence to underline its unique traits
Answer: It was corrected.
Line 173 – there is “Podosphera leucotricha”and should be “Podosphaera leucotricha”
Answer: Many Thanks for the correction, it was changed.
Line 254 – results presented in reference 32 you cited are not only Polish cultivars. Authors of the cited paper examined several old cultivars GROWN in Poland, not Polish ones
Answer: Thank you for the corection, it was changed.
Table 2 – are you sure that pectin and FRAP can be defined as biologically active compounds? Please check it carefully.
Answer: Thank you, you are right, the title was changed to „The health promoting properties of the old, local apple cultivars evaluated in experimental trials”
Line 521,534 – I am not sure if apples can contain amount of, or can be a source of FRAP, please check it carefully
Answer: Based on literature data, the apples have high water-soluble antioxidant capacity (FRAP).
Line 845-849 – please give more precise information on method that were used to determine optimal harvest time. Your description is too general. Please also describe methods of assessing each parameter in the same order as you present it in the Results section.
Answer: The sequence was changed. Some data were added about the optimal harvest time. The optimal harvest time was that time, which was suitable for storage (based on SSC and TTA, as well as organoleptic examination).
Thank you for your all help again. Have a great summer!
Yours sincerely,
the authors

Reviewer 4 Report
Comments and Suggestions for Authors
The authors have corrected their publication in accordance with the reviewer's suggestion. It is a pity, however, that they have not explained why they did not correct everything in accordance with the reviewer's opinion. Because it was requested that both parts of the variety name be written in capital letters. The authors did not do this in the context of the spelling of the varieties they studied. I further request that these changes be made.
Author Response
Dear Reviewer4,
Thank you for all of your time, efforts, help, which helped us a lot to improve the quality of this paper.
Here you can find our answer.
The authors have corrected their publication in accordance with the reviewer's suggestion. It is a pity, however, that they have not explained why they did not correct everything in accordance with the reviewer's opinion. Because it was requested that both parts of the variety name be written in capital letters. The authors did not do this in the context of the spelling of the varieties they studied. I further request that these changes be made.
Answer: Many Thanks for your suggestion, which helped a lot to improve quality of the paper. The cultivar names are very old names, and well-known in the Carpathian Basin, therefore we kept the original ones wirtten in the first two references (Lippay (1667, Bereczki M. (1887) – both are great Hungarian pomologists). The Hungarian pomologists use these cultivar names based on activity of Lippay and Bereczki today too.
Thank you for your all help again. Have a great summer!
Yours sincerly,
the authors
